# Rare and Undiagnosed Disease: A Learning Program for Nurses and Midwives

**DOI:** 10.3390/nursrep15050136

**Published:** 2025-04-22

**Authors:** Sue Baker, Kaila Stevens, Dale Pugh

**Affiliations:** Perth Children’s Hospital Rare Care Centre, Nedlands 6009, Australia; sue.baker@health.wa.gov.au (S.B.); kaila.stevens@health.wa.gov.au (K.S.)

**Keywords:** nursing, midwifery, rare disease, undiagnosed disease, learning program

## Abstract

This paper presents a newly developed online learning program currently designed to meet the learning objectives of nurses and midwives and rare and undiagnosed disease. **Background/Objectives:** This paper will also introduce the Global Nursing Network for Rare Disease and its role and commitment in supporting nurses and midwives in the identification of rare disease and the delivery of appropriate care and interventions to care for people living with rare and undiagnosed disease. Globally, nurses and midwives are often the first healthcare provider a patient will engage with. Combined with the estimated 300 million living with a rare disease across the globe, nurses and midwives are well positioned when assessing patients to have adequate awareness and suspicion to consider the presence and impact of rare disease. To enable this awareness and knowledge to ensure timely assessment and referrals, specific knowledge is required. There is a current paucity of learning programs about rare and undiagnosed disease specifically for nurses and midwives. **Methods:** The proposed learning program comprises seven modules designed to address the learning needs of novice to expert nurses and midwives from across the globe. Increased knowledge will in turn increase awareness and confidence to inform decision-making for patients presenting with undiagnosed signs and symptoms by ‘thinking rare’. **Results:** The proposed learning program comprises seven modules and a number of individual lessons which will be suitable for the needs of novice to expert nurses and midwives from across the globe.

## 1. Introduction

People living with rare and undiagnosed diseases (RUDs) are a global health priority recognised by the United Nations Resolution on Rare Diseases [1] and the formation of the Global Network for Rare Diseases, a partnership between the World Health Organisation and Rare Disease International [2] via a memorandum of understanding [3]. There are approximately 7000 rare diseases affecting more than 300 million people worldwide [4]. There are approximately 7000 identified rare diseases, with over 70% of them being realised in childhood [5]. Nurses and midwives, as the largest healthcare workforce are often the main and sometimes the only care providers for individuals living with RUD. Nurses and midwives commonly and frequently interact with patients and families across the age spectrum in primary health and acute care settings and therefore have a role and responsibility in caring for people living with rare and undiagnosed disease. Currently, there are limited RUD educational resources related to RUD for nurses and midwives and no specialised international post-graduate qualifications for nurses in RUD. To address global needs, it is crucial to develop accessible and equitable nursing and midwifery education. Therefore, the rationale for developing this learning program lies in the need for nurses and midwives to have knowledge about the nature and scope of rare and undiagnosed, including types of rare disease, commonality in patient signs and symptoms, and how they might present the length of time and barriers associated with having a diagnosis made. Knowledge of RUD is imperative in RD diagnosis, to in turn inform interventions to ensure the patient receives follow-up care and services. In turn, the rationale for presenting this paper is to increase awareness within nursing and midwifery communities and invite participation in the learning program. This paper will also introduce the Global Nursing Network for Rare Diseases (GNNRD) and its commitment and action to develop a suitable learning program to enhance awareness and knowledge for nurses and midwives.

## 2. Learning Program Objectives

The primary objective of this learning program is to increase awareness and enhance knowledge of RUD, building global scale and equity in the delivery of best practice across low-, middle-, and high-income countries. The program aims to equip participants with a comprehensive understanding of RUD, enabling them to deliver high-quality care, facilitate access to clinical trials, and advocate for those affected by RUD.

## 3. Global Nursing Network for Rare Diseases

Formally launched in March 2023, following the inaugural roundtable ‘Connecting Nurses Globally–A Roundtable in Rare and Undiagnosed Disease’ held in Singapore. The Rare Care Centre, SingHealth Duke NUS Genomic Medicine Centre and Curtin, Singapore, co-hosted this two-day event, which saw 33 nurses representing 25 countries and eight global leaders in rare disease assemble to collaborate at a global level. With equity of representation from low-, middle-, and high-income countries from North America, Africa, Europe, Asia, and Oceania, the foundations of the GNNRD were formulated and the urgent need for a nursing rare and undiagnosed disease education course identified.

The GNNRD is a central global facilitator for communication, information exchange, expertise, and resources to support the development and implementation of comprehensive care and best nursing practice with the collective vision to help improve the lives of PLWRUD. The aim of the GNNRD is to empower nurses and midwives globally to build capacity, individually and collectively through leadership, advocacy, knowledge exchange, and skills in partnership with and for the benefit of people living with rare and undiagnosed disease (PLWRUD). The operations of the network are underpinned by equitable access and care, cultural safety, and responsiveness and partnering with PLWRUD. All nurses and midwives will care for multiple PLWRUD during their career but may not be aware of their clinical and psychosocial needs and challenges. A globally coordinated approach provides the best opportunity to raise awareness and address the individual and collectively common challenges experienced by PLWRUD. The GNNRD is strategically positioning nurses to influence local, national, and global policy and advocacy platforms.

The development of a comprehensive learning program was recognised by the GNNRD Leadership Committee. The idea of a specific nursing learning program beyond the goals of creating awareness and informing and educating nurses is to ultimately enhance clinical outcomes for the patient and support patients and families to live their best life possible.

## 4. Learning Program Position Statement

To guide the development of a learning program, the GNNRD developed a position statement for the proposed education and learning program. The position statement provides to members and others guidance on the importance and required construct of education and learning programs for nurses and midwives who practice in rare and undiagnosed disease contexts and/or care for people and their families living with a RUD. The GNNRD asserts the following:Nurses and midwives and these professions have a privileged and imperative role in the care and support of all peoples, but this care for specific patient cohorts, that is, people living with a rare or undiagnosed disease, requires its own evidence base and related education to enable safe and quality care.While rare diseases are individually rare, they are collectively common, with over 350 million people across the world believed to have a rare disease. For this primary reason, the GNNRD asserts that all nurses and midwives require a degree of knowledge about rare and undiagnosed diseases.This knowledge should commence in undergraduate nursing and midwifery learning programs to inform early appreciation of the significance of understanding RUD and that all nurses and midwives have a role in the spectrum of care and services for PLWRD.The GNNRD, as a global leader for the practice of nursing and midwifery for people with a rare or undiagnosed disease, including clinical, management, education, research and policy roles, has a pivotal role in articulating the scope of practice and required knowledge and skills.

## 5. Education and Learning Committee

An education and learning committee has been established within the GNNRD. The inaugural committee of six members is responsible for the following:Providing leadership and expertise to contribute to the achievement of the strategic objective ‘Education and Learning’.Contributing to strategic planning for the Strategic Domain ‘Education and Learning’.Contributing to the development of curriculum for the learning program.Overseeing the development of the learning program and other activities.Making recommendations to the GNNRD Leadership Committee regarding learning programs, and ongoing reviews.Providing advice regarding topics and presenters for other education and learning activities, including but not limited to the Webinar and Symposium series.

## 6. What Do Nurses and Midwives Need to Know?

Few reports in the literature were identified that specifically addressed the learning needs of nurses and midwives. Despite this, the need for general and specialist nurses to have awareness and understanding of RUD as it relates to the patients and clients for a relevant scope of practice is recognised; see, for example, [6]. It has been asserted that although resources related to RUD are available, awareness of them is limited, and there is a need for education [7].

One study was conducted to identify the perceived genetic knowledge and education needs for this group of health professionals. Allied health professionals, nurses, and midwives participated. Findings related to identified topics reflecting learning needs were included: genetic fundamentals, genetic conditions specific to practice, understanding genetic testing, when and how to refer to genetic services, psychosocial implications, current evidence and research, ethical implications, and understanding professional roles within genetics [7]. In a more recent study conducted in Poland [8], 85% of 142 nurses and 75% of 113 student nurses agreed that there is a need for university courses to include rare diseases in the curricula, while 75% of nurses and 85% of student nurses expressed concern regarding how well prepared they felt to deal with patients with rare diseases. The internet was the major source of information related to rare diseases for the study participants.

Awareness and knowledge of RUD have applicability across all ages of the life continuum, from planning for pregnancy to antepartum and postpartum care, neonatology, infancy, childhood and adolescence, and adulthood. People living with a rare or undiagnosed disease will present to the community, primary care, acute care, and tertiary care settings for issues related to their RUD and other clinical and psychosocial reasons and needs. Therefore, these areas and contexts of practice are relevant to the intended nursing and midwifery participants of the learning program and modules.

The proposed content for the learning program was identified during the GNNRD Roundtable meeting held in Singapore with expert nurses. The following were identified and proposed for inclusion in the curriculum: genetics and genomics, physiology and pathophysiology (RUD ‘Red Flags’), genetic counselling, mental health support, trauma-informed care, patient/family centred care, care coordination best practice models, clinical trials and new treatments, and ethical considerations and critical analysis of research.

## 7. The Learning Program

The GNNRD has validated the need for and provided the international platform to advocate for and coordinate the development of a globally applicable course. A proposal for the learning program was developed and presented to the GNNRD members who attended the 2nd Annual Meeting of the Network in Abu Dhabi in 2024. Approval of the high-level approach and the proposed modules was obtained.

The program will include seven modules aimed at addressing different learning needs, from introductory to more comprehensive and specialised. These modules are presented in the following framework (Figure 1). Module 1 is being finalised in conjunction with Medics for Rare Disease (M4RD), who already have a number of introductory modules for health professionals [9]. Modules 1, 2, and 3 have been developed and are anticipated to be launched later this year. Planning is underway for Modules 4 to 7. The framework is complimented with the inclusion of the developmental stages of Benner’s Novice to Expert model for nurses and midwives wanting to align their nursing needs with the curriculum and subsequent modules [10].

The GNNRD contracted a specialist nurse educator to develop the content of Module 2 and Module 3 in conjunction with members of the Education and Learning Committee. The content was identified from a search of the contemporaneous literature related to RUD and, where available, nursing and midwifery practice and from RUD organisations. The review of content was conducted by one of the authors, with governance being provided by the Education and Learning Committee and the Leadership Committee. Eight lessons comprise Module 2:Care Coordination and Navigation in Rare Disease Nursing Practice.Communication and Relationship Building in Nursing Care for People Living with RUD.Mental Health Support in Rare Disease Care.Informing and Innovating Policy and Practice.Ethical Issues in Rare and Undiagnosed Diseases.Nursing and Midwifery Roles in Rare Disease Care.Introduction to Specific Diseases in RUD.Research and Clinical Trials.

Module 3 includes 18 diseases, including amyloidosis, alopecia, aplastic anaemia, Angelman Syndrome, Charcot–Marie–Tooth disease, childhood dementia, Fabry disease, fragile X syndrome, mitochondrial diseases, myasthenia gravis, sickle cell anaemia, and stiff-person syndrome. Each rare disease description provides a definition, prevalence, signs and symptoms, diagnostic considerations, known treatment or therapeutic interventions, and patient and family organisations and communities.

To ensure the modules meet high standards and the needs of learners, we have and will continue to engage expert reviewers for each lesson in Modules 1 and 2. Of the 24 lessons, 12 have at least two reviewers, 5 have one reviewer, and we are identifying additional reviewers for the remaining lessons. Feedback received so far varies, with some reviewers affirming the content’s adequacy and others offering valuable expansions, references, and links. No concerns have been raised regarding content accuracy, with suggestions focused on enhancing the content. The content has been subsequently modified to include these suggestions; lessons in Module 2 and 3 have been developed using a lesson plan with learning objectives, content including lived experience and videos, and assessment tasks. The program is being made accessible via the GNNRD web-based platform. This will enable participants to undertake the learning modules at their own pace. We anticipate that approximately six months will be needed to fully engage in all the learning activities for each module, although a participant will not be penalised if they require longer. The modules are initially being provided in English, but in keeping with our principle of equitable access, there is a plan for the modules to be presented in other languages. The first two languages that the program will translated to are Hindi and Portuguese. Participants will be awarded a certificate of completion upon attaining a pass in each lesson in each module.

Modules 4, 5, 6, and 7 will be in-depth modules for nurses and midwives wanting more information on care coordination, genetics and genomics, clinical trials, and mental health and wellbeing. Participants can choose to undertake one or all modules. Nurses or midwives may already be working in care coordination positions and wish to develop knowledge or seek in-depth knowledge to practice in the research area of clinical trials. Nurses and midwives may have a role in discussing and advising patients and families on genetic and genomic aspects of rare disease and diagnostic testing. The module will present genetic and genomic curricula relevant to nursing and midwifery practice. The psychosocial impact of rare and undiagnosed disease, impact on mental health, and the need for trauma-informed care will inform the lessons presented in Module 7.

All nurses and midwives and students of both the professions worldwide are invited to undertake the learning program. Access to the learning platform can be received after joining the GNNRD (gnnrd.org), which is at no cost to the individual. Further, there is no cost to undertaking the modules. The GNNRD is committed to ensuring that ability to afford education is not a barrier to learning.

## 8. Evaluation

Once the initial curriculum was designed for the two modules and individual lessons, it was evaluated by two cohorts of senior nurses with knowledge and experience of RUD, with sub-specialty knowledge in clinical practice and education. Participants will be asked to evaluate each lesson upon completion and provide feedback to the Education and Learning Committee. This feedback will inform the ongoing evaluation of the modules to ensure the lessons are easily interpreted, learning occurs, and the lessons are enjoyable. A pre- and post-test tool will be employed to assist the participant with meeting their learning objectives and provide the Education and Learning Committee with further information on how effective the lessons are helping the participant learn. Content will be evaluated at yearly intervals to ensure information is contemporaneous. By identifying strengths and areas for improvement, we plan to refine the curriculum to ensure that the learning needs of nurses and midwives continue to be met and a quality product is delivered.

## 9. Conclusions

All people deserve equitable access to healthcare. This equity includes the adequate awareness from all healthcare professionals when they are caring for and assessing patients who are yet to be diagnosed or have a rare disease. As part of a global organisation for nurses and midwives, we are translating the need for equitable and best practice healthcare with this innovative and much-needed learning program. Nurses and midwives are well positioned to meet the needs of PLWRUD. Awareness and understanding of the nature and associated presentations of rare and undiagnosed disease and the importance of early diagnosis, care coordination, and referrals are imperative to meeting the clinical and psychosocial needs of patients and families.

## Figures and Tables

**Figure 1 nursrep-15-00136-f001:**
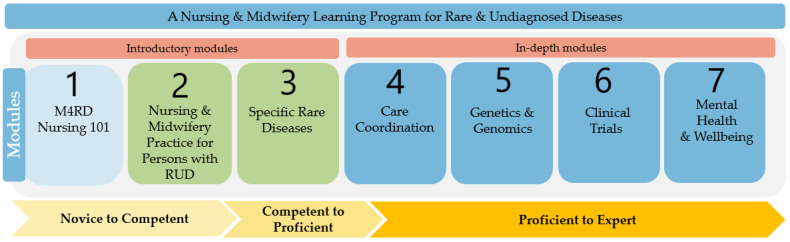
Learning program framework.

## Data Availability

Data sharing is not applicable. No new data were created or analyzed in this study.

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
