# Peer review of "Rare and Undiagnosed Disease: A Learning Program for Nurses and Midwives"

_nursrep, 2025, doi:10.3390/nursrep15050136_

Round 1
Reviewer 1 Report
Comments and Suggestions for Authors
I can say that the proposed educational programme is very suitable for the training of nurses and midwives.But it is too limited to be evaluated as research.
My suggestion to the researchers is to evaluate the feasibility of implementing the educational programme and to evaluate the competencies of nurses and midwives before and after the programme.
Author Response
Response to Reviewer 1 comments |
||
1. Summary |
|
|
Thank you very much for taking the time to review this manuscript. Please find the detailed responses below and the corresponding revisions/corrections highlighted and in track changes in the re-submitted files. |
||
3. Point-by-point response to Comments and Suggestions for Authors |
||
Comment 1: I can say that the proposed educational programme is very suitable for the training of nurses and midwives. But it is too limited to be evaluated as research. My suggestion to the researchers is to evaluate the feasibility of implementing the educational programme and to evaluate the competencies of nurses and midwives before and after the programme. |
||
Response 1: Thank you for pointing this out. We note that this paper is a ‘Communication’ paper and not a purely ‘Research’ paper. We are aiming to develop, implement, evaluate and re-evaluate content, design and effectiveness. Nonetheless we agree that more information is needed to better communicate this.
As such we have added the following:
Evaluation Once the initial curriculum was designed for the two modules and individual lessons, it was evaluated by two cohorts of senior nurses with knowledge and experience of RUD, with sub-specialty knowledge in clinical practice and education. Participants will be asked to evaluate each lesson upon completion and provide feedback to the Education and Learning Committee. This feedback will inform the ongoing evaluation of the modules to ensure the lessons are easily interpreted, learning occurs, and the lessons are enjoyable. A pre and post-test tool will be employed to assist the participant with meeting their learning objectives, and provide the Education and Learning Committee with further information on how effective the lessons are helping the participant learn. Content will be evaluated at yearly intervals to ensure information is contemporaneous. By identifying strengths and areas for improvement, we plan to refine the curriculum to ensure that the learning needs of nurses and midwives continue to be met, and a quality product is delivered.
This has been added to the manuscript using track changes.
|
Reviewer 2 Report
Comments and Suggestions for Authors
please see attached report

Author Response
Response to Reviewer 2 comments
|
||
1. Summary |
|
|
Thank you very much for taking the time to review this manuscript. Please find the detailed responses below and the corresponding revisions/corrections highlighted and in track changes in the re-submitted files. |
||
2. Point-by-point response to Comments and Suggestions for Authors |
||
Comment 1: T The UK curriculum requires that nurses are able to work across diverse care settings, including community care, primary care, and acute care, but the proposed GNNRD learning program does not explicitly address these different contexts. How will the authors address clinical practice and patient centred care across various settings?
|
||
Response 1: Thank you. This is a point that we had not considered in this paper, although we acknowledge that the need for awareness and knowledge about rare and undiagnosed goes across the ‘age continuum’, in fact from planning a pregnancy, neonates, infancy, childhood, adolescents, and adult hood’. As such, we have added this to the manuscript to make clear ‘the depth and breadth’ of practice where RUD knowledge is necessary.
Awareness and knowledge of RUD has applicability across all ages of the life continuum, from planning for pregnancy, antepartum and postpartum care, neonatology, infancy, childhood and adolescence, and adulthood. People living with a rare or undiagnosed disease will present to the following settings community, primary care, acute care, and tertiary care for issues related to their RUD, and other clinical and psycho-social reasons and needs. Therefore, these areas and contexts of practice have relevance to the intended nursing and midwifery participants of the learning program and modules. |
||
Comment 2: Whilst the GNNRD’s global approach allows for knowledge-sharing across low-, middle-, and high-income countries, what is the emphasise on evidence-based practice, since the GNNRD proposal lacks clarity on how research findings on RUD will be incorporated into education. How do the authors aim to address this issue as this is vital in determining clinical practice? |
||
Response 2: Thank you. Within the lessons, and where relevant, there is inclusion of evidence-based practice, noting that this is minimal in relation to actual nursing and midwifery practice. We note though, that there is not a lot of empirical evidence specifically related to the nursing or midwifery practice of and for those with a RUD. This identified gap will receive the attention of the GNNRD ‘Research and Innovation’ Committee which is in the process of being formed.
|
||
Comment 3: The UK curriculum requires nurses to recognize and respond to health inequalities, but the GNNRD framework does not fully address the socioeconomic barriers to diagnosis and treatment in rare diseases. How do the authors foresee these issues being addressed in different settings and countries? |
||
Response 3: Thank you. The GNNRD and the authors fully acknowledge this significant issue of inequity of access, diagnosis and care. The purposeful setting up of this global network, only commenced in 2023 has a clear mandate to enable and empower nurses and midwives to be better prepared and equipped to care for those living with a RUD. Notably we are not requiring participants to pay for the course. Philanthropic funding of nurses from across the globe to attend three meetings has occurred. We are also enabling language translation with Wordly, allowing those who do not speak English to more equitably participate in meetings, noting that this is being extended to education sessions.
The challenges of socio-economic barriers while not lost on us, are addressed in part by a highly committed and motivated RUD community of practice, including recent the recent position of the UN with a resolution of RD, and the work of the World Health Organisation, and other organisations and activities, including Hackathons.
|
||
Comment 4: Whilst the GNNRD learning program supports ongoing professional development, it does not address how it will specify collaboration with other healthcare professionals. How do the authors aim to address this issue? |
||
Response 4: Thank you. At this time the GNNRD are focused on developing our goals to meet the needs of nurses and midwives which was the initial and primary goal of the Network. The GNNRD and the authors have discussed extending not only membership to other health professionals, and those with lived experience, but also access to the learning programs. This is not completely off the table, and will be re-considered as we develop the membership and activities and attract ongoing funding to ensure all of our goals are robust and sustainable. |
||
Comments 5: Whilst the UK framework encourages nurses to take leadership roles in healthcare policy, the GNNRD should include more advocacy training to help nurses influence health policies related to RUD |
||
Response 5: Thank you. One of the main objectives of the GNNRD and its planned activities is to raise awareness for people living with RUD, and educate and empower nurses and midwives to advocate for these people, and for RUD in their own right – including earlier diagnosis, specific treatment and nuanced care and support.
Advocacy is included as a role of the Nurse in the ‘Communication of Practice’ Lesson, a Lesson on “Informing and Innovating for Policy’ has relevance to advocacy. The lesson on Nursing and Midwifery Roles includes advocacy.
We note this point and will take this back to the GNNRD Education and Learning Committee for further discussion, noting that we are considering developing a mentoring program, have recently welcome a student nurse member to the Education and Learning Committee and are seeking partnerships and opportunities to advocate for RUD and those living with a RUD.
|
||
Comment 6: Nurses in particular lack training in critical analysis of literature, and this has to be emphasised upon in the GNNRD proposal, as this then provides skills for a strong foundation in application of science to underpin the clinical practice. As noted in your paper, this needs to start at the UG level and be extended across any PG level education degree programs, which should also comprise of a dense component of experiential learning |
||
Response 6: Thank you. The authors do not disagree with this point. While the GNNRD is not in a position to be the source of education and learning opportunities for the many areas of knowledge and practice that nurses and midwives need, there is likely to be consideration of a learning package and /or resources for nurses and midwives as considered and decided by the GNNRD Research and Innovation Committee.
We believe that this point is out of scope for this paper. |
Reviewer 3 Report
Comments and Suggestions for Authors
Thank you for the opportunity to review the manuscript.
Please find below my comments
- The title is not clear and should be rephrased
- The abstract is too short and does not properly represent the study
- The introduction part should be expanded to provide the study rationale.
- The overall article is worth publishing after revision.
Thank you for the opportunity to review the manuscript.
Please find below my comments
- The title is not clear and should be rephrased
- The abstract is too short and does not properly represent the study
- The introduction part should be expanded to provide the study rationale.
- The overall article is worth publishing after revision.
Author Response
Response to Reviewer 3 comments
|
||
1. Summary |
|
|
Thank you very much for taking the time to review this manuscript. Please find the detailed responses below and the corresponding revisions/corrections highlighted and in track changes in the re-submitted files. |
||
3. Point-by-point response to Comments and Suggestions for Authors |
||
Comment 1: The title is not clear and should be rephrased
|
||
Response 1: Thank you. We have changed the title to ‘Rare and Undiagnosed Disease: A Learning Program for Nurses and Midwives’.
|
||
Comment 2: The abstract is too short and does not properly represent the study.
|
||
Response 2:
Thank you. We have re-written the Abstract to represent the totality of the paper more fully.
Abstract: This paper presents a newly developed online learning program currently designed to meet the learning objectives of nurses and midwives and rare and undiagnosed disease. This paper will also introduce the Global Nursing Network for Rare Disease and their role and commitment in supporting nurses and midwives in the identification of rare disease and the delivery of appropriate care and interventions to care for people living with rare and undiagnosed disease. Background/Objectives: Globally, nurses and midwives are often the first healthcare provider a patient will engage with. Combined with the estimated 300 million living with a rare disease across the globe, nurses and midwives are well positioned when assessing patients to have adequate awareness and suspicion to consider the presence and impact of rare disease. To enable this awareness and knowledge to ensure timely assessment and referrals specific knowledge is required. There is a current paucity of learning programs about rare and undiagnosed disease specifically for nurses and midwives. The proposed learning program comprises seven modules designed to address the learning needs of novice to expert nurses and midwives from across the globe. Increased knowledge will in turn increase awareness, and confidence to inform decision-making for patients presenting with undiagnosed signs and symptoms by ‘thinking rare’. The proposed learning program comprises seven modules and a number of individual lessons which will be suitable for the needs of novice to expert nurses and midwives from across the globe. |
||
Comment 3: The introduction part should be expanded to provide the study rationale. |
||
Response 3: Thank you. While this paper is aimed at communicating this learning program, rather than being a study in its own right, we have added the following to reflect the rationale of the program, and rationale for sharing this information.
The rationale for developing this learning program lies in the need for nurses and midwives to have knowledge about the nature and scope of rare and undiagnosed, including types of rare disease, commonality in patient signs and symptoms, and how they might present, the length of time and barriers associated with having a diagnosis made. Knowledge of RUD is imperative to have an awareness of the need to think that the patient may have a RD, and then to intervene to ensure that the patient receives follow-up care and services. In turn, the rationale for presenting this paper is to increase awareness within nursing and midwifery communities and invite participation in the learning program. |
||
Comment 4: The overall article is worth publishing after revision |
||
Response 4: Thank you. |
Reviewer 4 Report
Comments and Suggestions for Authors
Attend to the attached document with comments.

Author Response
Response to Reviewer 4 comments
|
||
1. Summary |
|
|
Thank you very much for taking the time to review this manuscript. Please find the detailed responses below and the corresponding revisions/corrections highlighted and in track changes in the re-submitted files. |
||
3. Point-by-point response to Comments and Suggestions for Authors |
||
Comment 1: Kindly indicate the program's primary objectives at the beginning of the paper after the introduction, as they were mentioned late in the middle.
|
||
Response 1: Thank you. We have moved the objectives with the heading ‘Learning Program Objectives’ after the ‘Introduction’.
Learning Program Objectives The primary objective of this learning program is to increase awareness and enhance knowledge of RUD, building global scale and equity in the delivery of best practice across low-, middle- and high-income countries. The program aims to equip participants with a comprehensive understanding of RUD enabling them to deliver high-quality care, facilitate access to clinical trials and advocate for those affected by RUD.
|
||
Comment 2: Authors spoke of launching the second and the third modules; kindly indicate the launching of the first module. When did you launch it? |
||
Response 2: Unfortunately, there have been some recent delays and this will be launched at the same time of Module 2 and 3, later this year.
We have changed what is written in the manuscript using track changes to read-
Modules 1, 2 and 3 have been developed and are anticipated to be launched later this year;. Planning is underway for Modules 4 to 7. |
||
Comment 3: You mentioned that module 2 comprises eight lessons, but you mentioned only seven; kindly correct it.
|
||
Response 3: Thank you. We have added the eighth lesson, ‘Research and Clinical Trials’ to the manuscript using track changes.
|
||
Comment 4: It was indicated that the modules would be presented in English, and other languages would be considered. Can you mention the other languages that you are planning to include?
|
||
Response 4: Thank you. The first two languages will be Portuguese and Hindi. We have added the following to the manuscript using track changes to-
The first two languages that the program will translated to are Hindi and Portuguese.
|
||
Comment 5: You indicated that the participants would undertake the learning modules at their own pace. Kindly indicate the planned duration of the program. Is it a sixmonth or 12-month program? Kindly indicate it. |
||
Response 5: Thank you. We acknowledge that there will be variation in the time needed to undertake each module, but we have provided the following information. We further note that six months is not dissimilar to a university unit of study or some emerging nursing micro-credential courses (for example the Australian College of Nursing.
We anticipate that approximately six months will be needed to fully engage in all the learning activities for each module, although a participant will not be penalized if they require longer.
The above point has been added to the manuscript using track changes.
|
||
Comment 6: Kindly indicate if there would be any practical module to be undertaken to assess the psychomotor skills of participants before they are awarded the certificates. |
Round 2
Reviewer 1 Report
Comments and Suggestions for Authors
I would like to express my sincere gratitude for the clarification provided.
In your text, you assert that:
"Once the initial curriculum was designed for the two modules and individual lessons, it was evaluated by two cohorts of senior nurses with knowledge and experience of RUD, with sub-specialty knowledge in clinical practice and education"
My question is, what comments or assessments did these expert nurses make of the modules they reviewed?
Were they useful in making any improvements or modifications to the initial proposal?
Author Response
Comment 1
My question is, what comments or assessments did these expert nurses make of the modules they reviewed?
Were they useful in making any improvements or modifications to the initial proposal?
Response 1
Many thanks for this further comment. I have written a short paragraph to explain the feedback process.
To ensure the modules meet high standards and the needs of learners, we have, and will continue to engage expert reviewers for each lesson in Modules 1 and 2. Of the 24 lessons, 12 have at least two reviewers, five have one reviewer, and we are identifying additional reviewers for the remaining lessons. Feedback received so far varies, with some reviewers affirming the content’s adequacy and others offering valuable expansions, references, and links. No concerns have been raised regarding content accuracy, with suggestions focused on enhancing the content. The content has been subsequently modified to include these suggestions.